# Effects of a Web-Based Lifestyle Intervention on Physical Fitness and Health in Physically Inactive Adults: A Randomized Controlled Trial

**DOI:** 10.3390/healthcare11212847

**Published:** 2023-10-29

**Authors:** Judith Brame, Jan Kohl, Christoph Centner, Ramona Wurst, Reinhard Fuchs, Iris Tinsel, Phillip Maiwald, Urs A. Fichtner, Matthias Sehlbrede, Erik Farin-Glattacker, Albert Gollhofer, Daniel König

**Affiliations:** 1Department of Sport and Sport Science, University of Freiburg, 79102 Freiburg, Germany; judith.brame@sport.uni-freiburg.de (J.B.);; 2Praxisklinik Rennbahn, 4132 Muttenz, Switzerland; 3Institute of Medical Biometry and Statistics, Section of Health Care Research and Rehabilitation Research, Faculty of Medicine and Medical Center–University of Freiburg, 79106 Freiburg, Germany; 4Centre for Sport Science and University Sports, Department of Sport Science, Division for Nutrition, Exercise and Health, University of Vienna, 1150 Vienna, Austria; 5Faculty of Life Sciences, Department of Nutritional Sciences, Division for Nutrition, Exercise and Health, University of Vienna, 1090 Vienna, Austria

**Keywords:** lifestyle intervention, web-based, physical fitness, physical activity behavior, dietary behavior, physiological health outcomes, health, adults, randomized controlled trial

## Abstract

Web-based lifestyle interventions are a new area of health research. This randomized controlled trial evaluated the effectiveness of an interactive web-based health program on physical fitness and health. *N* = 189 healthy adults participated in a 12-week interactive (intervention) or non-interactive (control) web-based health program. The intervention provided a web-based lifestyle intervention to promote physical activity and fitness through individualized activities as part of a fully automated, multimodal health program. The control intervention included health information. Cardiorespiratory fitness measured as maximum oxygen uptake (VO_2_max) was the primary outcome, while musculoskeletal fitness, physical activity and dietary behavior, and physiological health outcomes were assessed as secondary outcomes (t0: 0 months, t1: 3 months, t2: 9 months, t3: 15 months). Statistical analysis was performed with robust linear mixed models. There were significant time effects in the primary outcome (VO_2_max) (t0–t1: *p* = 0.018) and individual secondary outcomes for the interactive web-based health program, but no significant interaction effects in any of the outcomes between the interactive and non-interactive web-based health program. This study did not demonstrate the effectiveness of an interactive compared with a non-interactive web-based health program in physically inactive adults. Future research should further develop the evidence on web-based lifestyle interventions.

## 1. Introduction

Physical inactivity is one of the most important risk factors for noncommunicable diseases (NCDs) [1]. The latest global estimates show that physical inactivity accounts for up to 8% of NCDs [2]. Globally, 27.5% of the adult population (23.4% of men and 31.7% of women) is not sufficiently physically active [3] and does not meet current physical activity guidelines [4]. With such a worldwide prevalence of physical inactivity, nearly 500 million new cases of preventable NCDs are expected by 2030, resulting in a cost of USD 300 billion (INT 524 billion) to health systems [5].

However, the global burden of physical inactivity can be addressed through the wide-ranging health-promoting potential of a physically active lifestyle. The health benefits of physical activity, as well as of physical fitness as a component of physical activity, have been well documented for both the prevention and treatment of many NCDs, such as obesity, type 2 diabetes mellitus, and metabolic syndrome, and for individual related health outcomes, such as anthropometry, blood pressure, blood levels, and vascular health [6,7,8]. Thus, lifestyle interventions should include a focus on physical activity, as recommended in clinical guidelines [9,10,11].

Previous research has shown that face-to-face interventions can be successful in managing NCDs [12,13,14]. As this has been extensively researched, and as various information and communication technologies have developed in our digital age, opening up additional opportunities for healthcare delivery, web-based lifestyle interventions have become a new focus of research [15]. From a public health perspective, delivering interventions via the internet offers potential advantages over traditional face-to-face interventions. For example, web-based interventions can reach large populations at low costs [16]. The number of internet users worldwide has been increasing over the past few decades and was estimated to be around 5.4 billion by 2023 [17]. Furthermore, the ubiquitous accessibility and availability of web-based interventions can increase user convenience [16], and interventions can be individually tailored and interactive through multiple technological design options [18,19]. To fully realize this potential, scientific research on the effectiveness of web-based lifestyle interventions is warranted. Systematic reviews and meta-analyses of web-based lifestyle interventions to promote physical activity or fitness [20,21,22,23,24] and to prevent and treat NCDs [25,26,27,28,29,30] suggest beneficial effects but still point to several research gaps, particularly studies on long-term effects and objectively measured health outcomes. Further research is consequently needed to strengthen the indicative evidence on web-based lifestyle interventions.

This study evaluated a web-based lifestyle intervention promoting physical activity and fitness as part of a multimodal health program offered by a national statutory health insurance fund (Techniker Krankenkasse, TK). This 12-week interactive web-based health program (“TK-HealthCoach”, TK-HC) aims to achieve prioritized health goals (e.g., “Increasing Fitness”, “Losing and Maintaining Weight”, or “Smoking Cessation”) through individualized health coaching for primary prevention of NCDs [31]. The present randomized controlled trial focused on the evaluation of the health goal “Increasing Fitness” with objectively measured health outcomes from medical examinations. These medical examinations were carried out in southwest Germany by the Department of Sport and Sport Science at the University of Freiburg (DSS) [32].

The objective of this study was to investigate the effectiveness of a 12-week interactive web-based health program (TK-HC) on physical fitness and health in physically inactive adults. The program was compared with a 12-week non-interactive web-based health program. For this investigation, we defined three hypotheses focusing on long-term effects. Short- and medium-term effects were additionally considered. We hypothesized that the interactive web-based health program would significantly improve primary and secondary outcomes with small to medium effect sizes (time effect). We further postulated that these improvements would be more significant than those of the non-interactive web-based health program (interaction effect). Moreover, we expected that higher program use of the interactive web-based health program would result in better program effectiveness than lower program use (dose–response relationship) [32].

## 2. Materials and Methods

### 2.1. Study Design

This study was part of a nationwide evaluation project of the TK-HC, in which the health goal “Increasing Fitness” was evaluated in a randomized controlled online trial and the present randomized controlled trial. The online trial assessed self-reported outcomes from online questionnaires of participants across Germany and was conducted by the Section of Health Care Research and Rehabilitation Research at the Medical Center of the University of Freiburg (SEVERA) [33]. In addition, the present trial assessed objectively measured health outcomes from medical examinations of participants living only in southwest Germany. These medical examinations occurred at four measurement time points (t0: 0 months, t1: 3 months, t2: 9 months, t3: 15 months) at the DSS. The study was approved by the Ethics Committee of the University of Freiburg (237/19, 25 July 2019) and registered at the WHO-approved German Clinical Trials Register (https://drks.de, DRKS00020249, accessed on 11 December 2019). The study protocol has already been published [32]. This article reports the study results using the CONSORT-EHEALTH checklist [34].

### 2.2. Participants

To register for the study, subjects had to be participants of the online trial living in southwest Germany (postcode area: 79). Male, female, and diverse adults (18–65 years) were included if they had a body mass index (BMI) between 18.5 and 34.9 kg/m^2^ and a physical activity level of ≤60 min/week. They also had to be willing to be randomized and predominantly healthy. If subjects had any health problems or diseases, they were asked to have a medical certificate for study participation. Internet literacy was not specified as an inclusion criterion but was strongly recommended. Pregnancy was a criterion for exclusion [32].

Subjects were recruited through various online and offline media, including Google advertisements, websites, newsletters, and social media, as well as local press, radio, and flyers. All recruitment tools led to an open-access landing page where interested subjects were provided with study information and asked to give their written informed consent by ticking a box. The study information and consent form could then be downloaded. The information provided in these documents can be viewed together with the published study protocol. Upon study registration, participants were randomized and contacted by study staff to receive additional study information, assess eligibility, and arrange an appointment for the first medical examination. As part of the first medical examination, participants were reinformed, reassessed against the inclusion and exclusion criteria, and given the opportunity to discuss any remaining concerns. Participants were finally enrolled in the study after successful completion of the first medical examination. The face-to-face assessments during the medical examinations on-site allowed the study staff to get to know the participants over time. Further information on the enrollment process, particularly in relation to the online trial, can be found in the study protocol [32].

### 2.3. Interventions

#### 2.3.1. Interactive Web-Based Health Program

The program assigned to the intervention group was the TK-HC. This fully automated, web-based health program provides interactive, multimodal health coaching tailored to personal health goals and profiles. Based on motivational psychology, users are expected to achieve sustainable lifestyle changes in a self-directed and flexible way within 12 weeks [31]. As the program offers different coaching modules, this study focused on the “TK-FitnessCoaching” (TK-FC). This module promotes physical activity and fitness through endurance, strength, flexibility, and coordination training [32].

To start the program, users had to pass a multistep anamnesis and were then guided to the program platform, which was structured into three areas. In the first area (“My health program”), users could access a personal dashboard with a weekly and an overall 12-week coaching plan of individual activities (Figure 1A). Users could choose these activities mainly from 13 fitness activities to promote physical activity and fitness in their daily lives (Figure 1B). These fitness activities (e.g., “I improve my endurance by running” or “I strengthen my muscles according to a recommended training plan”) centered on endurance and strength training but also included flexibility and coordination training and other health-related fitness courses. All fitness activities came with detailed workout instructions and could be practiced at home, in a gym, or outdoors. The activities were conducted over a 12-week training period, and the number of weekly training sessions varied by the activity. Endurance and strength activities required a regular standardized fitness test. In addition to these predefined fitness activities, other self-determined sports activities could be carried out [32].

Furthermore, users were allowed to choose activities from other coaching modules, such as weight loss activities from the “TK-WeightLossCoaching” (TK-WC), and connect an activity tracker to the program to send their daily step count. Users were asked to log all their activities on a daily or weekly basis and could flexibly adjust them to their individual needs. In addition to the activities, the dashboard included a barrier management tool, customized tips and notifications, support options, and a visualization of the current personal health goal achievement [32].

The second area of the program (“Knowledge”) assisted users with evidence-based background information about physical activity and fitness and other health-related topics. The contents were taught through interactive knowledge courses prepared in small articles with texts, videos, tools, and tests. One course (“Optimal training”) was automatically scheduled weekly in the coaching plan. In the third area (“My data”), users could find an analysis profile of their logged activities, their health profile and personal data, as well as the study information and consent form. Moreover, they could withdraw participation from the study and the health program. Once users had completed the program, they received a coaching summary [32].

#### 2.3.2. Non-Interactive Web-Based Health Program

The control group was offered a 12-week non-interactive web-based health program. This program was a simple online platform designed to help users promote their physical activity and fitness. The program initially required users to log in without providing any anamnesis in order to tailor the health program to personal health goals or profiles. The first area (“My health program”) provided only non-interactive health information on physical activity and fitness. Users could select and read short articles from standardized texts, but these articles did not include interactive elements such as videos, tools, or tests. In contrast to the interactive web-based health program, users also did not have access to health information on other health goals, for example, weight loss. There were also no daily or weekly activities to promote their physical activity and fitness, and they could not link an activity tracker to the program. In the second area (“My data”), users could view their personal data, the study information, and the consent form and could withdraw participation from the study and the health program. The program did not comprise an individual analysis profile or user feedback mechanisms [32].

Both interventions were scientifically developed by TK in collaboration with health professionals and Vilua Healthcare GmbH (VHG). The interventions were accessible as desktop and mobile versions for unlimited use on computers, tablets, and smartphones and were delivered as web-based interventions without human involvement. A support team from VHG was available by phone or e-mail to answer questions about the content and technical aspects of the health programs as needed. No instructions or recommendations were given to users on program use. The interactive web-based health program (TK-HC) was tested and revised in advance, including usability tests and two randomized controlled pilot trials. The program was then frozen and used throughout the study period without any changes in software development that could have influenced the results [32]. As the program is owned by TK and only accessible to members of the health insurance fund outside the study setting, only screenshots of the program can be made publicly available (Figure 1A,B).

After completing the first medical examination, participants were given access to the interactive or non-interactive web-based health program. They were authorized to log in to the respective health program for free via the project’s landing page using a personal study account. During the study period, participants received regular e-mail reminders to participate in the health programs, which could be run as often as desired until the end of the study. Participants were asked not to take part in any other physical activity or fitness studies. For successfully completing the study, participants received free medical examinations, an activity tracker (Fitbit Charge 3^TM^) (San Francisco, CA, USA) [35], free access to the current version of the TK-HC, and an individual study report. More details of the interventions are provided in the study protocol [32].

### 2.4. Outcomes

#### 2.4.1. Primary Outcome

##### Cardiorespiratory Fitness

To evaluate health goal achievement, we defined physical fitness, specifically cardiorespiratory fitness, as the primary outcome, which was assessed by maximum oxygen uptake (VO_2_max) (mL/min/kg) using the Cooper 12-min run test [36]. VO_2_max was calculated from the maximum distance (m) achieved on a 400 m running track with the formula “$\dot{V}$O_2max_ (mL·kg^−1^·min^−1^) = (distance in meters − 504.9)/44.73” [37] (p. 118). Heart rate (220 − age ± 10 bpm) and the Borg Rating of Perceived Exertion (RPE) scale (≥17) served as criteria for exertion. Participants were provided with a standardized warm-up and cool-down program and a test meal [32].

#### 2.4.2. Secondary Outcomes

##### Musculoskeletal Fitness

As another component of physical fitness, musculoskeletal fitness was determined as a secondary outcome. It was evaluated by maximum isometric strength (N, kg) with an isometric leg press [38,39] and a hand grip [40]. For the leg press test, participants were laying in a standardized supine position with a 90° knee and hip angle, and after a submaximal familiarization phase (4 repetitions, 50–60% of maximum voluntary contraction (MVC): 2–3 s, rest: 1 min), bilateral maximum isometric strength (N) was assessed (3 trials, 100% MVC: 2–3 s, rest: 3 min). For the hand grip test, participants used a standardized sitting, arm, and grip position, and maximum isometric strength (kg) was taken alternately on the left and right hand (3 trials, 100% MVC: 2–3 s). Participants were required to complete a standardized warm-up before strength testing. In both procedures, the maximum of three successful trials was scored [32].

##### Physical Activity and Dietary Behavior

Behavioral health outcomes were further included as secondary outcomes. Physical activity was analyzed by daily step count (steps/day), sedentary behavior (min/day), and light, moderate, and vigorous intensity physical activity (min/week) via the Fitbit Charge 3^TM^ [35] and the International Physical Activity Questionnaire (long version) (IPAQ-L) [41]. Dietary behavior was monitored for energy intake (kcal/day), nutrient intake (g/day), and food consumption (Healthy Eating Index of the German National Nutrition Survey, HEI-NVS) (score) [42,43] by 7-day dietary records with NutriGuide^®Plus^ software (version 4.8) [44]. As part of the first medical examination, participants were instructed on how to use the Fitbit Charge 3^TM^ and dietary records [32].

##### Physiological health outcomes

Moreover, secondary physiological health outcomes were obtained. For anthropometry, body weight (kg), body height (m), BMI (kg/m^2^), fat mass and fat-free mass (kg), and waist circumference (cm) were measured standardized with a stadiometer (seca 274) [45], a bioelectrical impedance analysis scale (seca mBCA 515) [46], and a measuring tape (seca 201) [47]. Waist circumference was assessed at the midpoint between the lowest rib and the iliac crest, using the mean of two successful measurements. Prior to measurement, participants were advised to use the toilet and to take off all clothing except underwear and any accessories [32].

In addition, systolic and diastolic blood pressure (mmHg) and heart rate (bpm) were recorded with an electronic blood pressure device [48] on the left upper arm of the participants in a standardized sitting, arm, and cuff position. Two successful measurements were taken at 1–2 min, from which the mean was calculated. The measurement included a preceding rest phase, was conducted in a closed and quiet room, and participants were asked to be relaxed. Furthermore, venous blood sampling was used for the analysis of blood glucose (fasting plasma glucose (mg/dL), glycated hemoglobin (HbA_1c_) (%)) and blood lipids (triglycerides, total cholesterol, high-density lipoprotein (HDL) cholesterol, low-density lipoprotein (LDL) cholesterol (mg/dL)) [32].

As a final outcome, vascular health was assessed in terms of endothelial function by flow-mediated dilatation (FMD) (%) with the non-invasive AngioDefender^TM^ system [49]. The 8-min measurement was performed as the blood pressure assessment, but only one measurement was taken. Participants’ vascular age (days) was estimated by Vascular Age Calculator^TM^ (VAC) software (version 4.0.8) [50] with the use of FMD and physiological health outcomes [32].

All outcomes were collected at four medical examinations (t0: 0 months, t1: 3 months, t2: 9 months, t3: 15 months) (approximately 4–5 h each) with standardized procedures by trained and qualified staff in separated and protected premises of the DSS. For all medical examinations, participants had to appear fasting (meals and drinks, nicotine: ≥12 h, alcohol and physical exercise: ≥48 h) with sufficient sleep the night before and in good health. This was asked at each medical examination within an anamnesis completed by the study director, which included documentation of diseases, medication, and adverse events. Data were discussed with the principal medical investigator and used to review the inclusion and exclusion criteria and further study participation. Participants were thoroughly introduced to each examination. If they could not fulfill the examination criteria or complete an examination as required, the examination was repeated. There were no specific risks associated with the medical examinations or the health programs in advance. The availability of the principal medical investigator ensured that immediate care could be taken on-site if needed. He could also stop participation if any health problems arose that precluded further participation in the study. Additional information on all primary and secondary outcomes can be taken from the study protocol [32].

#### 2.4.3. Other Outcomes

Usage data of the interactive web-based health program were monitored during the intervention and follow-up phase (logins/week) to analyze the effect of program use on program effectiveness. Data were assessed by VHG and sent to the DSS [32].

### 2.5. Sample Size

Sample size calculation was performed by SEVERA with the use of G*Power software (version 3.1) [51] for the primary outcome (VO_2_max). The calculation required a case number of *n* = 79 for each study group and, with an additional dropout rate of 15%, a total number of *n* = 186 (93 + 93). Recruitment activities were stopped when this total sample size was reached [32,33].

### 2.6. Randomization

Randomization was performed as permuted block randomization with variable block sizes of 4, 6, and 8 and an allocation ratio of 1:1. Randomization sequences were prepared by SEVERA with RITA software (version 1.50) [52] and transferred to VHG for secure storage to conceal group allocation until the interventions were assigned. This guaranteed that randomization was done without any influence. After successful study registration on the project’s landing page, participants were automatically randomized by computer to the study groups based on the computer-generated randomization lists [32,33].

### 2.7. Blinding

Participants were told about the intervention character of the interactive and non-interactive web-based health program as part of the study information for ethical reasons. They were then able to identify the allocated health program after their first program login and therefore could not be blinded. Blinding of recruiters and outcome assessors was possible by providing information on group allocation only after completion of all medical examinations. In addition, participants were advised not to disclose their group allocation to outcome assessors during the medical examinations on-site. However, it was possible for them to do so. Data monitors and analysts could not be blinded because conclusions about group allocation could be drawn from the data [32,33].

### 2.8. Statistical Methods

Data analysis was performed with R software (version 4.0.5) [53]. Primary and secondary outcomes were analyzed by robust linear mixed models using the R package robustlmm [54]. The significance level was set at 0.05. Cohen’s *d* was calculated as the effect size with a 95% CI (|*d*| = 0.2: small effect, |*d*| = 0.5: medium effect, |*d*| = 0.8: large effect) [55]. Outcomes were examined in a per-protocol (PP) and an intention-to-treat (ITT) analysis. The PP analysis included only participants who attended all four medical examinations and had complete data for the relevant outcome (complete cases). The ITT analysis included all randomized participants. Missing values were estimated by multiple imputation using the R package micemd [56]. As the PP and ITT analyses covered many outcomes and provided similar results, the present analysis is limited to the ITT analysis. For the analysis of usage data, latent class analysis was used to define different user types based on their weekly login behavior. Due to the different and small sample sizes of each user type, data were analyzed descriptively [32].

### 2.9. Data Management

Data collection and storage were done in pseudonymized form and publication in aggregated form. Personal data were kept secure and separate from research data. There was no data transfer to third parties. Deletion of all personal data will occur after three years, and deletion of all research data after ten years. Participants took part in the study and the health programs voluntarily and could withdraw participation anytime through the health programs, the DSS, or VHG. They were briefed on all privacy issues in the study information and gave their written informed consent. In case of withdrawal, the consent form and personal data were deleted, and research data were stored anonymously. Further details are described in the study protocol [32].

## 3. Results

### 3.1. Participants

Participants were recruited between January and September 2020. A total of *N* = 318 subjects registered for the study, of which *n* = 189 were finally included in the medical examinations. The medical examinations took place from January 2020 to December 2021. Thus, the study was conducted during the coronavirus disease 2019 (COVID-19) pandemic. Throughout the study period, *n* = 83 dropouts and no intervention-related serious adverse events were reported. As an additional *n* = 10 participants paused individual medical examinations due to the COVID-19 pandemic, *n* = 96 participants were included in the PP analysis. Fewer subjects were considered if missing data were available for the relevant outcome. For the ITT analysis, all *n* = 189 participants were analyzed. The detailed flow of participants is visualized in Figure 2. Demographic and anthropometric characteristics of the participants are described in Table 1. The following presented results of the primary and secondary outcomes refer to the ITT analysis.

### 3.2. Primary Outcome

#### Cardiorespiratory Fitness

The change in VO_2_max in both study groups over the course of the study is illustrated in Figure 3A. From t0 to t3, VO_2_max improved from 25.94 ± 6.29 to 27.52 ± 5.38 mL/min/kg (+1.58 mL/min/kg, +6.09%) in the intervention group and from 26.64 ± 5.77 to 27.80 ± 4.96 mL/min/kg (+1.16 mL/min/kg, +4.35%) in the control group (Table 2). Statistical analysis showed a significant short-term time effect with a small effect size for the intervention group (t0–t1: *p* = 0.018, *d* = 0.33) (Table 3 and Table 4). There were no other significant time or interaction effects (Table 4).

### 3.3. Secondary Outcomes

#### 3.3.1. Musculoskeletal Fitness

Figure 3B shows the change in maximum isometric strength as measured by the leg press test. The intervention group improved their maximum isometric strength from 1261.85 ± 473.62 to 1303.67 ± 342.26 N (+41.82 N, +3.31%) and the control group from 1274.11 ± 441.60 to 1364.77 ± 319.69 N (+90.66 N, +7.12%) (t0–t3) (Table 2). No significant time or interaction effects were found (Table 3 and Table 4). There were also no significant time or interaction effects for maximum isometric strength as assessed by the hand grip test (Table 2, Table 3 and Table 4).

#### 3.3.2. Physical Activity and Dietary Behavior

In terms of physical activity behavior, the intervention group increased their daily step count from t0 to t3 from 10,298.98 ± 3275.36 to 10,342.08 ± 2726.89 steps/day (+43.10 steps/day, +0.42%) and the control group from 10,110.05 ± 2934.87 to 10,425.40 ± 3059.40 steps/day (+315.35 steps/day, +3.12%). In addition, total physical activity decreased from 2276.06 ± 572.77 to 2259.61 ± 511.09 min/week (−16.45 min/week, −0.72%) in the intervention group and increased from 2229.46 ± 423.58 to 2257.68 ± 423.92 min/week (+28.22 min/week, +1.27%) (t0-t3) in the control group (Appendix A). These changes tracked with the Fitbit Charge 3^TM^ are presented in Figure 3C,D. Statistical analysis revealed no significant time or interaction effects for these outcomes or for sedentary behavior and light, moderate, and vigorous intensity physical activity (Appendix A). For physical activity as reported by the IPAQ-L, a significant short- and medium-term time effect with small to medium effect sizes was observed for the intervention group for vigorous intensity physical activity (t0–t1: *p* < 0.001, *d* = 0.52; t0–t2: *p* = 0.008, *d* = 0.47). No other significant time or interaction effects were found (Appendix A). Regarding dietary behavior, a significant medium-term time effect with a small effect size for the intervention group was seen for alcohol intake (t0–t2: *p* = 0.039, *d* = −0.20). Otherwise, no significant time or interaction effects were detected (Appendix A).

#### 3.3.3. Physiological Health Outcomes

Statistical analysis of the physiological health outcomes showed a significant long-term time effect with a small effect size for the intervention group for waist circumference (t0–t3: *p* = 0.039, *d* = −0.27). For diastolic blood pressure, there was a significant medium-term time effect with a small effect size for the intervention group (t0–t2: *p* = 0.039, *d* = 0.22) and a significant medium-term interaction effect in favor of the control group (t0–t2: *p* = 0.023) (Appendix A). However, a significant group difference was observed at t0 (*p* = 0.023), so this interaction effect was not considered as given. Furthermore, a significant medium- and long-term time effect with very small to small effect sizes for triglycerides (t0–t2: *p* = 0.035, *d* = −0.15; t0–t3: *p* = 0.026, *d* = −0.23) and a significant short- and medium-term time effect with small effect sizes for LDL cholesterol (t0–t1: *p* = 0.034, *d* = 0.28; t0–t2: *p* = 0.035, *d* = 0.34) were found for the intervention group (Appendix A). There were no other significant time or interaction effects for the physiological health outcomes (Appendix A).

### 3.4. Other Outcomes

For the analysis of usage data, all participants of the intervention group (*n* = 104) were classified into the following user types: regular users (*n* = 58, 55.80%), half-time users (*n* = 13, 12.50%), partial users (*n* = 13, 12.50%), rare users (*n* = 16, 15.40%), and one-time users (*n* = 4, 3.80%). Figure 4 presents the change in VO_2_max over the course of the study in each type of user compared to the control group. From t0 to t3, data did not indicate a consistent dose–response relationship (regular users: +1.49 mL/min/kg (+6.02%), half-time users: +0.96 mL/min/kg (+3.56%), partial users: +1.52 mL/min/kg (+5.68%), rare users: +1.79 mL/min/kg (+6.19%), one-time users: +4.29 mL/min/kg (+17.32%), control group: +1.16 mL/min/kg (+4.35%)) (Table 5).

## 4. Discussion

This study aimed to investigate the effectiveness of a 12-week interactive web-based health program (TK-HC) on physical fitness and health in physically inactive adults. For this purpose, the program was compared with a 12-week non-interactive web-based health program. The effectiveness of the interactive web-based health program in terms of superiority over the non-interactive web-based health program was not proven. Significant time effects were found in the primary outcome and individual secondary outcomes for the interactive web-based health program, but there were no significant interaction effects in any of the primary or secondary outcomes between the interactive and non-interactive web-based health program. There was also no consistent dose–response relationship between different user types of the interactive web-based health program.

Web-based lifestyle interventions have emerged as an attractive approach to the prevention and treatment of NCDs due to their compelling advantages [16,18,19]. Moreover, since the COVID-19 pandemic, they have become even more important as a home-based tool to keep people physically active [57], particularly as physical inactivity has been associated with an increased risk of severe COVID-19 outcomes [58]. In the present study, an interactive web-based health program to promote physical activity and fitness was not superior to a non-interactive web-based health program. To contextualize our findings within the existing literature, systematic reviews and meta-analyses on this topic of web-based interventions focus primarily on physical activity and include only a few studies on physical fitness, especially cardiorespiratory fitness, and physiological health outcomes. In addition, the underlying heterogeneity of the studies limits the ability to compare web-based interventions in terms of content and methodology. Nevertheless, these investigations suggest positive short- and medium-term effects with small to medium effect sizes of web-based interventions [20,21,22,23,24]. This could not be seen in our study results.

Looking at individual studies in more detail, with respect to cardiorespiratory fitness as our primary outcome, the study by Hansen et al. [59] investigated a web-based intervention on physical activity and health measurements in physically inactive adults. The intervention group (*n* = 6055) received a physical activity website, and the control group (*n* = 6232) was not exposed to any intervention. A subsample (*n* = 1190) of these participants, who were part of a nationwide survey, was also invited to health examinations. After three months, there were no significant improvements in cardiorespiratory fitness assessed by a watt-max or sub-max test on a bicycle ergometer (−0.40 mL/min/kg vs. +0.30 mL/min/kg) or musculoskeletal fitness assessed by a hand grip test (+0.60 kg vs. −2.40 kg) in the intervention and control group. Additionally, no significant improvements in BMI (−0.10 kg/m^2^ vs. ±0.00 kg/m^2^), waist circumference (−0.10 cm vs. −0.50 cm), fat mass (±0.00% vs. ±0.00%), or systolic (+0.10 mmHg vs. ±0.00 mmHg) and diastolic blood pressure (−0.30 mmHg vs. −0.40 mmHg) were observed in both study groups. Our results were almost consistent with this study and revealed only a significant time effect in cardiorespiratory fitness in the intervention group (see Section 3.2) but also no significant improvements in musculoskeletal fitness, BMI, waist circumference, fat mass, or systolic and diastolic blood pressure in the intervention and control group after three months (see Section 3.3).

Regarding physical activity as one of our secondary outcomes, the study by Compernolle et al. [60] examined a web-based, computer-tailored, pedometer-based physical activity intervention in working adults. The intervention group (*n* = 137) was provided with a pedometer, an information booklet, and computer-tailored step advice, while the control group (*n* = 137) was given no intervention. After three months, pedometer-based daily step count increased significantly by 1064.91 steps/day in the intervention group, compared with a decrease of 24.72 steps/day in the control group. Our results showed a non-significant increase of steps/day measured by the Fitbit Charge 3^TM^ in the intervention and control group after three months (see Section 3.3). Another study by Wijsman et al. [61] evaluated a web-based intervention on physical activity and metabolic health in inactive older adults. The intervention group (*n* = 119) was offered a web-based physical activity program (Philips DirectLife), and the control group (*n* = 116) was assigned to a waiting list. After three months, there was a significant increase in daily moderate-to-vigorous intensity physical activity, as measured by a wrist accelerometer, of 11.10 min/day in the intervention group, compared with a decrease of 0.10 min/day in the control group. Moreover, body weight (−1.49 kg vs. −0.82 kg), waist circumference (−2.33 cm vs. −1.29 cm), fat mass (−0.64% vs. +0.07%), and HbA_1c_ (−0.05% vs. −0.01%) improved significantly more in the intervention group than in the control group. In our study, there was a non-significant decrease in weekly moderate and vigorous intensity physical activity measured by the Fitbit Charge 3^TM^ in the intervention and control group after three months. Furthermore, improvements in body weight, waist circumference, fat mass, and HbA_1c_ were also not significant in both study groups after three months (see Section 3.3).

Although our study could not confirm most of the findings in the literature, our study was not limited to short- and medium-term effects but specifically focused on long-term effects as a predefined research gap. Furthermore, many studies on web-based interventions use self-reported measures of physical activity, which are susceptible to systematic overestimation [62]. Our study comprised an elaborate assessment of objectively measured health outcomes, including cardiorespiratory and musculoskeletal fitness, physical activity and dietary behavior, and physiological health outcomes. However, as this study found no evidence of effectiveness, more research is needed on web-based lifestyle interventions promoting physical activity and fitness, especially on long-term effects and on cardiorespiratory and musculoskeletal fitness and physiological health outcomes.

Possible explanations for our study results may be found within the current scientific literature. Eysenbach [63] described two challenging issues in the evaluation of web-based interventions in a randomized controlled trial. On the one hand, it is difficult to ensure that the intervention is really being used by the intervention group, which can be checked by evaluating usage data in a dose–response analysis. In the present study, the interactive web-based health program provided an individually tailored health coaching with high complexity and diversity. Users could choose their preferred activities from a wide range of options to promote their physical activity and fitness. The program made recommendations about how much exercise to do to achieve health benefits, but users could not be required to follow them. Our analysis of usage data showed that *n* = 58 (55.50%) participants of the intervention group were classified as regular users, but there was no consistent dose–response relationship between the different categorized user types. As our analysis was based only on the weekly login behavior of the users, a more detailed analysis and, thus, a better control on the part of the study may have provided further information on whether participants were using the intervention appropriately, e.g., considering endurance and strength activities. In addition, our study participants had the opportunity to give open, qualitative feedback on the interactive and non-interactive web-based health program within each anamnesis, which was noted by the study director. Feedback from participants of our intervention group indicated that the interactive web-based health program may not have been sufficiently developed. Participants reported that the program was very well-founded and interesting but that its usability could be improved. The program was further perceived as complex and time-consuming. Participants also asked for more guidance within the program and more suitability for everyday use, e.g., through an app. Particularly during the COVID-19 pandemic, with high levels of personal and work-related stress, it was challenging for participants to use the program.

On the other hand, it is difficult in evaluating web-based interventions to ensure that the control group is really a control group and not using other web-based interventions out of disappointment at not receiving the intervention. It is therefore recommended to assess the use of other web-based interventions among control group participants [63]. Qualitative feedback from participants of our control group was indicative of this issue. Participants reported that they were disappointed at being in the control group and, therefore, created their own training plan, followed other web-based health programs, or used the Fitbit Charge 3^TM^ to achieve their personal health goals. However, this feedback was open and voluntary, so it should be treated with caution. A standardized evaluation of the activities of the control group participants may have provided more reliable information. Taken together, these two issues of the intervention and control group may have led to what is known as “bias towards the null” [63] (p. 2), making it difficult to show significant interaction effects between the interactive and non-interactive web-based health program in our study.

In addition to these difficulties, namely the two issues of the intervention and control group, this study had further methodological limitations that may have influenced the results. First, it should be noted that the study population was defined very broadly. However, it had to correspond to the target group of the interactive web-based health program. To assess potential influence of age and gender, additional adjusted PP and ITT analyses for the primary outcome (VO_2_max) were performed, which are not presented in this article. These analyses showed that age and gender did not affect the results. This study also had a high dropout rate, so the calculated sample size could not be maintained until the end of the study. The ITT analysis was performed to account for this. In addition, it was not possible to blind participants, which is common in eHealth trials [63,64].

Moreover, from a measurement perspective, our study participants may have been more motivated due to the face-to-face assessments on-site than participants of the online trial or other purely web-based trials. However, this was not seen in our data. Further, it was challenging to provide a feasible and standardized assessment of physical activity and fitness in our large sample size. The primary outcome (VO_2_max) was measured using the Cooper 12-min run test, which has been shown to be a valid field test for estimating cardiorespiratory fitness [36,65]. Musculoskeletal fitness was assessed by an isometric leg press and hand grip test as acceptable measures of maximum isometric strength [66,67]. However, it should be considered whether these tests were specific enough to reflect training adaptations [68], as strength activities of the interactive web-based health program focus on strength endurance training. Strength endurance tests were pretested for our strength assessment, but they were difficult to standardize. For physical activity, we used the Fitbit Charge 3^TM^ as a measurement tool for both study groups, but there is evidence that Fitbit-based interventions can increase physical activity behavior [69]. Despite this and mixed evidence on the validity and reliability of Fitbit devices [70,71], several studies used these devices as an intervention or measurement tool, including in the control group [72].

Finally, it needs to be mentioned that this study was conducted during the COVID-19 pandemic. Studies have shown that this pandemic had a negative impact on physical activity behavior worldwide [73]. As mentioned above, this was also reported by our study participants and was a reason for dropout. Nevertheless, web-based interventions have been considered beneficial as helpful measures to stay physically active in times of severe restrictions in daily life [74].

Taken together, these limitations may reduce the generalizability of our study results. Given the methodological approach and the difficulties encountered, our study could not provide evidence of the superiority of an interactive web-based health program over a non-interactive web-based health program. For future studies, we would like to suggest several recommendations. Studies evaluating web-based lifestyle interventions to promote physical activity and fitness should establish more pronounced differences between the intervention and control group. For the intervention group, it is essential to ensure that the intervention is used appropriately. To verify this, detailed usage data should be collected. For the control group, it should be considered whether a waiting list would be better than a control intervention, and the use of other web-based interventions should be evaluated. A high level of control of the intervention and control group with reliable data may provide helpful information to justify the study results. Furthermore, the study population should be narrowly defined to minimize variability in the data, and the study sample should be large enough to achieve the required sample size despite a high dropout rate. Attention should also be paid to the specificity of training and testing methods used to assess physical fitness. In addition, it is important to differentiate between the intervention itself and the measurement tool used to assess physical activity. With regard to the health program being evaluated, it is important that the program has been sufficiently developed and previously tested. On the one hand, it should offer a high level of usability, and on the other hand, it should guide the user through the program in a targeted and close way to achieve health-promoting effects. Overall, we definitely need more randomized controlled trials to evaluate the effectiveness of web-based lifestyle interventions.

## 5. Conclusions

This study examined a 12-week web-based lifestyle intervention to promote physical activity and fitness as part of a multimodal health program. The effectiveness of this interactive web-based health program in comparison with a non-interactive web-based health program on physical fitness and health in physically inactive adults was not demonstrated. However, this study included an elaborate assessment of objectively measured health outcomes and was part of a nationwide evaluation project during the COVID-19 pandemic. Further research is needed to extend the available evidence on web-based lifestyle interventions to promote physical activity and fitness.

## Figures and Tables

**Figure 1 healthcare-11-02847-f001:**
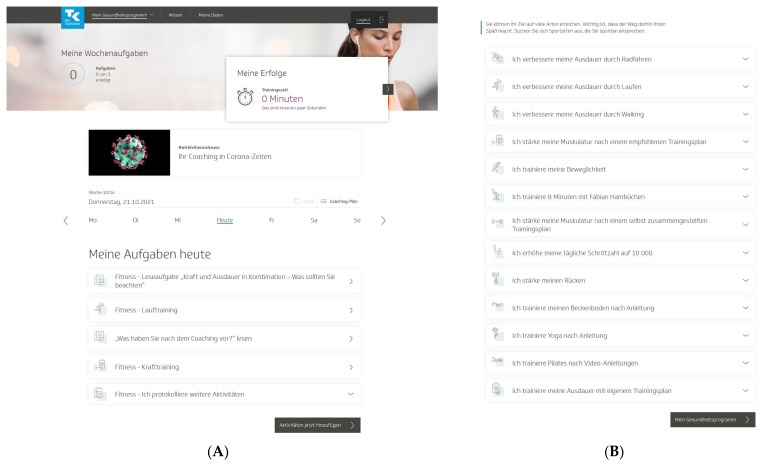
“TK-HealthCoach” (TK-HC): (**A**) personal dashboard and (**B**) fitness activities of the “TK-FitnessCoaching” (TK-FC). TK: Techniker Krankenkasse. Source: project’s landing page (https://gesundheitsprogramm.tk.de/de/studien/gesundheitsziele (accessed on 21 October 2021)).

**Figure 2 healthcare-11-02847-f002:**
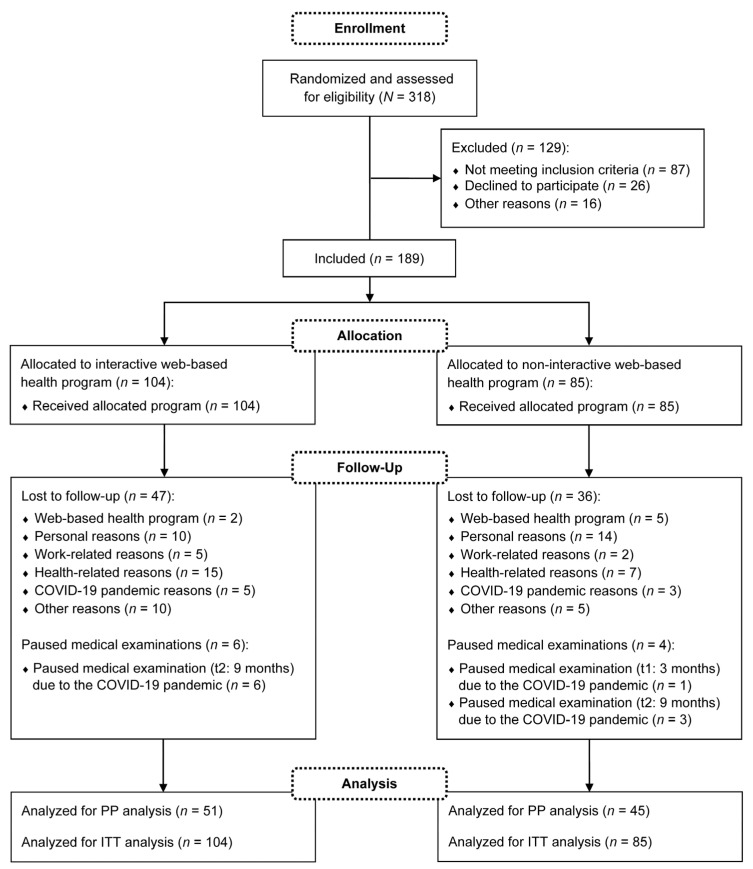
Flow of participants. COVID-19: coronavirus disease 2019; PP: per-protocol; ITT: intention-to-treat.

**Figure 3 healthcare-11-02847-f003:**
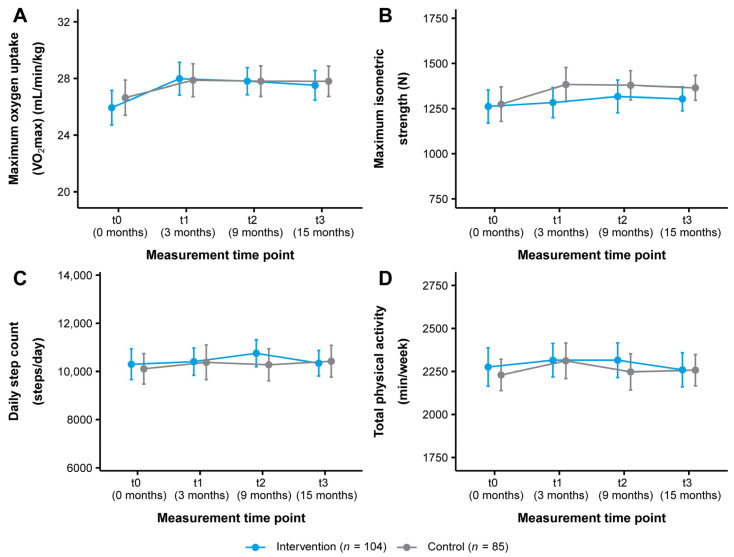
Change in (**A**) cardiorespiratory fitness, (**B**) musculoskeletal fitness, (**C**) daily step count, and (**D**) total physical activity in the intervention and control group (ITT analysis, *n* = 189). Data are presented as mean ± 95% CI. Maximum oxygen uptake (VO_2_max): Cooper 12-min run test, maximum isometric strength: leg press test, daily step count/total physical activity: Fitbit Charge 3^TM^. ITT: intention-to-treat.

**Figure 4 healthcare-11-02847-f004:**
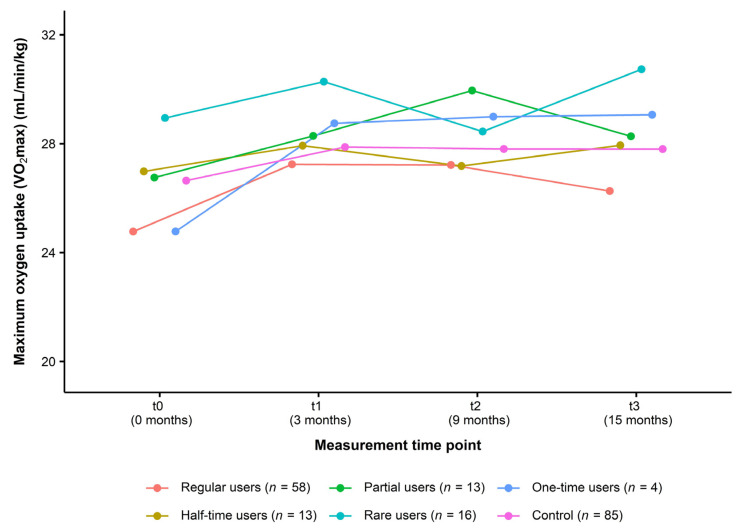
Change in cardiorespiratory fitness in different user types of the intervention group compared to the control group (Usage data of the ITT analysis, *n* = 189). Data are presented as mean. Maximum oxygen uptake (VO_2_max): Cooper 12-min run test. ITT: intention-to-treat.

**Table 1 healthcare-11-02847-t001:** Demographic and anthropometric characteristics of participants.

Variable	Total (*n* = 189)	Intervention (*n* = 104)	Control (*n* = 85)	*p*-Value
Age (years)	48.07 ± 12.01	48.36 ± 11.65	47.73 ± 12.49	0.722
Gender (n)				
Male	50 (26.50)	28 (26.90)	22 (25.90)	0.872
Female	139 (73.50)	76 (73.10)	63 (74.10)	
Body weight (kg)	74.65 ± 13.70	74.99 ± 13.01	74.24 ± 14.57	0.713
Body height (m)	1.70 ± 0.08	1.70 ± 0.08	1.71 ± 0.08	0.523
BMI (kg/m^2^)	25.56 ± 3.66	25.85 ± 3.50	25.19 ± 3.84	0.216

Data are presented as mean ± SD or frequencies (%). Statistical significance was tested using chi-squared test and independent *t*-tests. BMI: body mass index.

**Table 2 healthcare-11-02847-t002:** Descriptive statistics for physical fitness (ITT analysis, *n* = 189).

Group	t0 (0 Months)	t1 (3 Months)	t2 (9 Months)	t3 (15 Months)
Maximum oxygen uptake (VO_2_max) (mL/min/kg)				
Intervention	25.94 ± 6.29	27.98 ± 5.95	27.81 ± 4.94	27.52 ± 5.38
Control	26.64 ± 5.77	27.88 ± 5.37	27.80 ± 4.98	27.80 ± 4.96
Maximum isometric strength (leg press) (N)				
Intervention	1261.85 ± 473.62	1283.36 ± 430.61	1317.15 ± 466.86	1303.67 ± 342.26
Control	1274.11 ± 441.60	1382.98 ± 438.19	1378.98 ± 375.60	1364.77 ± 319.69
Maximum isometric strength (hand grip) (kg)				
Intervention	39.85 ± 11.24	39.21 ± 9.03	40.24 ± 9.17	40.57 ± 8.38
Control	39.58 ± 9.82	39.42 ± 9.11	41.08 ± 7.87	41.82 ± 8.14

Data are presented as mean ± SD. Intervention: *n* = 104, control: *n* = 85. ITT: intention-to-treat.

**Table 3 healthcare-11-02847-t003:** Effect sizes for physical fitness (ITT analysis, *n* = 189).

Group	t0–t1 (0–3 Months)	t0–t2 (0–9 Months)	t0–t3 (0–15 Months)
Maximum oxygen uptake (VO_2_max) (mL/min/kg)			
Intervention	0.33 [0.06, 0.61]	0.33 [0.06, 0.64]	0.27 [−0.00, 0.54]
Control	0.22 [−0.08, 0.52]	0.22 [−0.09, 0.52]	0.22 [−0.09, 0.52]
Maximum isometric strength (leg press) (N)			
Intervention	0.05 [−0.22, 0.32]	0.12 [−0.16, 0.39]	0.10 [−0.17, 0.37]
Control	0.25 [−0.06, 0.55]	0.26 [−0.05, 0.56]	0.24 [−0.07, 0.54]
Maximum isometric strength (hand grip) (kg)			
Intervention	−0.06 [−0.33, 0.21]	0.04 [−0.23, 0.31]	0.07 [−0.20, 0.35]
Control	−0.02 [−0.32, 0.28]	0.17 [−0.13, 0.47]	0.25 [−0.05, 0.55]

Data are presented as Cohen’s *d* [± 95% CI]. Interpretation: |*d*| = 0.2: small effect, |*d*| = 0.5: medium effect, |*d*| = 0.8: large effect. Intervention: *n* = 104, control: *n* = 85. ITT: intention-to-treat.

**Table 4 healthcare-11-02847-t004:** Results of robust linear mixed models for physical fitness (ITT analysis, *n* = 189).

Predictors	Maximum Oxygen Uptake (VO_2_max) (mL/min/kg)		Maximum Isometric Strength (Leg Press) (N)		Maximum Isometric Strength (Hand Grip) (kg)	
	Estimate	*p*-Value	Estimate	*p*-Value	Estimate	*p*-Value
(Intercept)	25.03 ± 1.36	<0.001 *	1206.50 ± 93.98	<0.001 *	37.59 ± 1.95	<0.001 *
Time						
t0–t1 (0–3 months)	2.65 ± 1.12	0.018 *	−18.70 ± 92.77	0.840	−0.32 ± 1.73	0.854
t0–t2 (0–9 months)	2.47 ± 1.28	0.055	1.35 ± 94.84	0.989	−0.27 ± 1.77	0.880
t0–t3 (0–15 months)	1.83 ± 1.54	0.236	34.09 ± 107.97	0.752	−0.00 ± 1.93	1.000
Group (control)	0.71 ± 0.88	0.425	11.20 ± 61.55	0.856	0.52 ± 1.27	0.681
Time * group (control)						
t0–t1 (0–3 months)	−0.70 ± 0.71	0.326	67.31 ± 59.95	0.262	0.05 ± 1.11	0.967
t0–t2 (0–9 months)	−0.70 ± 0.84	0.401	58.68 ± 61.99	0.344	0.82 ± 1.19	0.492
t0–t3 (0–15 months)	−0.29 ± 1.05	0.786	31.71 ± 69.54	0.649	1.20 ± 1.26	0.341

Data are presented as regression coefficient ± SE. Statistical significance (*p* < 0.05) is indicated (*). Intervention: *n* = 104, control: *n* = 85. ITT: intention-to-treat.

**Table 5 healthcare-11-02847-t005:** Descriptive statistics for physical fitness (Usage data of the ITT analysis, *n* = 189).

Group	t0 (0 Months)	t1 (3 Months)	t2 (9 Months)	t3 (15 Months)
Maximum oxygen uptake (VO_2_max) (mL/min/kg)				
Intervention				
Regular users	24.77 ± 5.30	27.24 ± 5.96	27.22 ± 5.01	26.26 ± 4.78
Half-time users	26.98 ± 6.31	27.93 ± 5.96	27.18 ± 5.37	27.94 ± 4.27
Partial users	26.75 ± 6.78	28.28 ± 7.22	29.95 ± 2.83	28.27 ± 7.64
Rare users	28.94 ± 8.26	30.28 ± 5.39	28.44 ± 5.93	30.73 ± 5.59
One-time users	24.77 ± 7.52	28.75 ± 2.65	28.99 ± 2.41	29.06 ± 2.01
Control	26.64 ± 5.77	27.88 ± 5.37	27.80 ± 4.98	27.80 ± 4.96

Data are presented as mean ± SD. Intervention: *n* = 104 (regular users: *n* = 58, half-time users: *n* = 13, partial users: *n* = 13, rare users: *n* = 16, one-time users: *n* = 4), control: *n* = 85. ITT: intention-to-treat.

## Data Availability

Research data from this study can only be accessed through this publication. Data supporting reported results are contained within this publication and the Appendix A. Data have only been published as aggregated research data so that individual data cannot be identified. All other research data will be retained by the DSS and will not be made publicly available.

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
