# Peer review of "Effects of a Web-Based Lifestyle Intervention on Physical Fitness and Health in Physically Inactive Adults: A Randomized Controlled Trial"

_healthcare, 2023, doi:10.3390/healthcare11212847_

Round 1
Reviewer 1 Report
Comments and Suggestions for Authors
The manuscript was very well-written and, in my opinion, very complete. I do not have many suggestions.
One suggestion was in the introduction/background (lines 57-66). I was not sold on the web-based lifestyle intervention being necessary. There is some evidence listed there but I was looking for more of a reasoning as to why web-based interventions are necessary compared to face-to-face.
Another suggestion was to describe the non-interactive intervention a bit more. In the methods section, I could not fully grasp what the difference was between the two and a little more clarification would help.
Other than these two, well-written, great work.
Reviewer 2 Report
Comments and Suggestions for Authors
The comments are provided in the PDF using comments and sticky notes.

Reviewer 3 Report
Comments and Suggestions for Authors
Thank you for the opportunity to review this study, my considerations are below:
Title:
- The title is confusing, if it only brings the primary outcome of the study, therefore, I suggest another title for the study: Effects of a 12-Week Web-Based Lifestyle Intervention on Cardiorespiratory and Physical Fitness, and Lifestyle-Related Health Outcomes in Physically Inactive Adults: A Randomized Controlled Trial.
Abstract:
- Did the control group participate in any type of intervention or continued carrying out their daily activities? This needs to be described in the abstract.
- I believe that the intervention and its characteristics should be better described by the authors. Using just the term used by the authors is too vague to know what this intervention was about.
- There is a loose sentence in the abstract: “Program use was also considered.”
- I suggest improving the text of the results, for example, the authors report improvements, however, they do not mention which outcomes. Furthermore, are these improvements intra-group, or in comparison between groups? This also needs to be described.
- Furthermore, I suggest including the names of the outcomes when mentioning that there was an improvement and the p-values of the outcomes, demonstrating that there was indeed a statistical difference in the data.
- The results are confusing, in the results the authors reveal that there were significant differences, in the conclusions they say that there were none.
- The abstract must be rewritten with great attention to the suggestions above.
Introduction:
- The introduction, despite being well written, does not make it clear why this type of intervention program is proposed. Why would people stop doing conventional training programs to do this? What would be the advantages? This needs to be included in the introduction.
- The objective at the end of the introduction is much better described than the objective of the abstract, which presents different elements. I suggest standardizing a single objective across the entire study.
Methods:
- I suggest standardizing the term “randomized controlled trial” throughout the study.
- Did the volunteers have illnesses? Which? This is very important, as the presence of diseases can influence the results of the study.
- Who carried out the randomization of the study? This information is important.
- Who carried out the anamnesis before the volunteers were directed to the intervention platform? What data were analyzed in these various stages of the anamnesis?
- Could users choose any of the 13 options? So we may have a bias there, as the volunteers may have opted for different training.
- How long was the training period for the volunteers to carry out the interventions? Was this time standardized by the researchers? Why not mention this in the study. Was there control over the regularity of volunteers in the sessions? How many sessions in total did the volunteers do during these 12 weeks? This information is very important and must be included in the manuscript.
- I suggest including photos of the intervention, so that readers can better understand how the intervention worked.
- I suggest that authors divide the outcomes into subtopics, for a better understanding of what was evaluated.
- The authors should better describe the process of blinding outcome assessors and data analysts, the current text is confusing.
Results:
- I suggest adding the p-value when comparing the sample characteristic data in table 1, showing that the samples are comparable and in the other tables.
- A big problem with this study is that the authors evaluated many outcomes and did not divide them to present their results. So, it becomes a tiring and long text for those who are reading it. I suggest that the authors divide and classify the outcomes into the methods, through a division into categories, demonstrating the results and discussion based on this division. For example: physical outcomes....cardiorespiratory outcomes.
Discussion:
- The discussion is the most “scientifically poor” part of the study.
- The authors do not discuss the findings, and initially bring up many concepts, which were previously mentioned in the introduction and repeat them in the discussion of the manuscript.
- The discussion is very vague and superficial, the authors only mention that they found results similar or not to those of other studies, this is not a discussion of the findings.
- I missed a physiological explanation of what may have happened to justify the results, because the exercises used did not promote improvements in the outcomes, this would bring greater theoretical-scientific solidity to the findings and especially to this type of intervention and its exercises.
- I suggest adding to the limitations, the methodological limitations of this study and emphasizing that its results cannot be generalized.
- The authors mention the data already described in the tables in the discussion. Authors can indicate in the text where to find the data, but mentioning them in the results and then citing them once again in the discussion is unnecessary.
- In the discussion, the authors mention that 55% of the intervention group were classified as regular. There is no mention in the methods of how this measurement occurred. And this could justify the results of the study.
- I missed the authors mentioning more about the difficulties encountered, and suggesting improvements to the instrument and indicating improvements and what the authors should control better, which was not possible in this study, for this type of intervention as a recommendation for future studies on the theme. For example, a scale integrated with the device, which every time the volunteer loses weight, provides words of encouragement and positivity. Or that the device provides the study authors with the weekly time that the volunteer did or did not do the intervention, is important data to improve this type of intervention in the future.
Conclusions:
- The final conclusion and the conclusion of the abstract are different, the authors must standardize on a single conclusion throughout the study. Furthermore, the authors should better analyze whether or not using the term showed significant differences in the study results.
Round 2
Reviewer 3 Report
Comments and Suggestions for Authors
Congratulations to the authors, for the changes in the work, I believe there are still a few changes missing, however, the scientific text has improved a lot.
The authors made significant changes to the manuscript, which greatly improved in terms of text clarity and scientific soundness. However, I believe that the title still needs to be modified, that is, there should not be any words after the term "Randomized Controlled Trial"... thus, that text that is after the term RCT should come before the colon.
Furthermore, the authors use the term "Web-Based Lifestyle Intervention" and this term and the description of the exercises is also very vague and leaves room for different interpretations. Therefore, I suggested including photos of the intervention, as I believe they better demonstrate what this type of intervention was like and increase the chances of this study being cited when using this intervention in future RCTs on this intervention. However, I believe that the photos should be included in the manuscript, it would add a lot to the study.
Given the above, I decided that the article should be accepted for publication, however with these two changes, therefore, I decided to accept after minor revisions.
